# Abbreviated Versus Multiparametric Prostate MRI in Active Surveillance for Prostate-Cancer Patients: Comparison of Accuracy and Clinical Utility as a Decisional Tool

**DOI:** 10.3390/diagnostics13040578

**Published:** 2023-02-04

**Authors:** Fabio Zattoni, Silvio Maresca, Fabrizio Dal Moro, Iliana Bednarova, Gianmarco Randazzo, Giovanni Basso, Giuseppe Reitano, Gianluca Giannarini, Chiara Zuiani, Rossano Girometti

**Affiliations:** 1Department Surgery, Oncology and Gastroenterology, Urologic Unit, University of Padova, 35122 Padova, Italy; 2Department of Medicine, Institute of Radiology, University of Udine, Santa Maria della Misericordia University Hospital, 33100 Udine, Italy; 3Department of Breast Radiology, Veneto Institute of Oncology, IRCCS, 35128 Padua, Italy; 4Urology Unit, Santa Maria della Misericordia University Hospital, 33100 Udine, Italy

**Keywords:** prostate MRI, prostate cancer, active surveillance, multiparametric MRI, abbreviated MRI protocol

## Abstract

(1) Purpose: To compare the diagnostic accuracy between full multiparametric contrast-enhanced prostate MRI (mpMRI) and abbreviated dual-sequence prostate MRI (dsMRI) in men with clinically significant prostate cancer (csPCa) who were candidates for active surveillance. (2) Materials and Methods: Fifty-four patients with a diagnosis of low-risk PCa in the previous 6 months had a mpMRI scan prior to a saturation biopsy and a subsequent MRI cognitive transperineal targeted biopsy (for PI-RADS ≥ 3 lesions). The dsMRI images were obtained from the mpMRI protocol. The images were selected by a study coordinator and assigned to two readers blinded to the biopsy results (R1 and R2). Inter-reader agreement for clinically significant cancer was evaluated with Cohen’s kappa. The dsMRI and mpMRI accuracy was calculated for each reader (R1 and R2). The clinical utility of the dsMRI and mpMRI was investigated with a decision-analysis model. (3) Results: The dsMRI sensitivity and specificity were 83.3%, 31.0%, 75.0%, and 23.8%, respectively, for R1 and R2. The mpMRI sensitivity and specificity were 91.7%, 31.0%, 83.3%, and 23.8%, respectively, for R1 and R2. The inter-reader agreement for the detection of csPCa was moderate (k = 0.53) and good (k = 0.63) for dsMRI and mpMRI, respectively. The AUC values for the dsMRI were 0.77 and 0.62 for the R1 and R2, respectively. The AUC values for the mpMRI were 0.79 and 0.66 for R1 and R2, respectively. No AUC differences were found between the two MRI protocols. At any risk threshold, the mpMRI showed a higher net benefit than the dsMRI for both R1 and R2. (4) Conclusions: The dsMRI and mpMRI showed similar diagnostic accuracy for csPCa in male candidates for active surveillance.

## 1. Introduction

Multiparametric magnetic resonance imaging (mpMRI) stratifies the risk of clinically significant prostate cancer (csPCa, Gleason ≥ 3 + 4) and monitors patients enrolled in active surveillance (AS) programs [1,2]. After an initial PCa diagnosis, confirmatory prostate biopsies may be recommended to better identify otherwise occult csPCa [3,4]. However, it is known from the Promis study that mpMRI has advantages over systematic biopsies for the detection of csPCa. After a positive mpMRI in the AS setting, there are higher risks of reclassification and upgrading before a confirmatory biopsy and a radical prostatectomy, respectively. According to a systematic review by Schoots et al. [5], the reclassification rate before starting an AS protocol is higher in patients who undergo mpMRI (39%) than in those who do not (17%). This suggests that MRI is helpful in guiding targeted biopsies of lesions not otherwise suitable for AS and, at the same time, in reducing patient overtreatment. Notably, it is estimated that about 11% of patients have csPCa with a negative MRI. This suggests the misleading risk assessment (under grading and volume underassessment) of systematic biopsy and MRI [6].

The minimum technical standards for mpMRI were part of both versions of the Prostate Imaging Reporting and Data System (PI-RADS) scoring systems [7,8]. In each case, the standard examination was defined with diffusion-weighted imaging (DWI), T2-weighted imaging (T2WI), and dynamic contrast-enhanced imaging (DCE) sequences. However, there is an intense debate over whether the abbreviated magnetic resonance imaging (MRI) protocols can achieve comparable diagnostic accuracy and biopsy guidance for csPCa while avoiding mpMRI-related limitations, such as direct and indirect costs, long acquisition time, and contrast-agent side-effects [9,10].

Different abbreviated MRI protocols have been investigated, with various numbers and types of sequences acquired compared to the standard composition of mpMRI [9]. Dual-sequence MRI (dsMRI), including transverse T2-weighted imaging and DWI only, is the minimum feasible protocol. This approach was tested on biopsy-naive patients or patients with negative biopsy history, with good results in terms of accuracy [11,12,13]. However, no previous studies explored the role of abbreviated protocols in patient candidates for AS before confirmatory biopsy. Within this scenario, the impact of comparable results between MRI images with and without contrast agents would be to shorten MRI examinations, improve patients’ acceptance of and compliance with the AS protocol, and increase prostate MRI availability.

The aim of the present study was thus to compare the accuracies of dsMRI and mpMRI in assessing csPCa in patients initially referred for AS.

## 2. Materials and Methods

### 2.1. Patient Selection and Study Design

This retrospective single-center study was approved by the institutional review board (approval number prot. IRB 29_2020)**.**

The acquisition of informed consent was waived because of the retrospective design.

The following inclusion criteria were used for patients with a mpMRI performed in our institution from February 2015 to November 2018:Diagnosis of low-risk PCa in accordance with the National Comprehensive Cancer Network NCCN classification [14].Candidates for mpMRI cognitively-guided transperineal targeted biopsy for index lesions (PI-RADS ≥ 3) and concomitant transperinal saturation biopsy.A mpMRI performed before the saturation biopsy and 3–12 months after the initial 12 core biopsy diagnostic for PCa.

We excluded patients with:

Time from diagnosis of PCa to mpMRI exam exceeding the established temporal cut-off.Incomplete mpMRI protocol due to contraindications to contrast medium administration.Presence of artifacts affecting imaging quality.Incomplete clinical data.A mpMRI performed before the first standard biopsy.Any previous prostate surgery.

### 2.2. MRI Protocol

A 3.0T MRI scanner (Achieva, Philips Medical Systems) with a 32-channel surface coil was used. All patients were premedicated with a cleansing enema and the administration of a spasmolytic drug (20 mg. i.m. of hyoscine butylbromide (Buscopan, Boehringer Ingelheim)).

The mpMRI consisted of triplanar T2WI, DWI, and DCE imaging; the latter was obtained after i.v. administration of 0.1 mmol/Kg of gadoteridol (Prohance, Bracco) or gadoteric acid (Dotarem, Guerbet), followed by 20 mL of saline flush, at an injection rate of 3 mL/s. Appendix A summarizes the acquisition protocol. The DWI was acquired as a single sequence, using the b = 1000 s/mm^2^ images to build the apparent diffusion coefficient (ADC) map with a monoexponential model fitting signal decay versus b-values. All the b-values were natively acquired. The DCE images were obtained in both native and subtracted forms.

The dsMRI was extrapolated from mpMRI and consisted of transverse T2WI and DWI. The latter included maximum b-value images (2000 s/mm^2^) and the ADC map.

### 2.3. Image Analysis

Image analysis was performed independently by two radiologists, namely reader 1 (R1) and reader 2 (R2). According to the MRI-quality-assessment criteria [15], R1 was classed as “experienced” (expertise of >1000 mpMRI), while R2 was qualified a “basic” prostate radiologist (expertise of 500–1000 mpMRI). Radiologists were blinded to biopsy results. In order to reflect the real-world scenario, radiologists were aware that mpMRI examinations had been performed prior to AS confirmatory biopsy. During each reading session, a study coordinator assigned to each radiologist the mpMRI and the dsMRI of the same patient at different times. To avoid recall bias, mpMRI and dsMRI of the same patient were examined in different reading sessions four weeks apart by each reader.

Readers were required to use the PI-RADS v2.1 classification [16]. Since dsMRI did not include DCE, we assumed that observations of the peripheral zone (PZ) categorized as PI-RADS 3 could not be upgraded. Each mpMRI index lesion was recorded by its position on the PI-RADS v2.1 sector map. Findings of PI-RADS ≥ 3 were considered positive.

### 2.4. Standard of Reference and Data Analysis

Patients had a mpMRI cognitive-guided transperineal biopsy for every finding of PI-RADS ≥ 3, followed by an institutional standardized 24-core-saturation prostate biopsy. Random transperineal-biopsy protocol were performed with 24-core (ten cores in the peripheral zone and two cores in the transitional zone in each prostatic lobe, respectively) according to a previously published scheme [17]. Histological analysis was performed by one of three genitourinary pathologists with experience ranging from 5 to 20 years. Categorical variables were reported as frequency, while continuous variables were reported as median and interquartile range.

The sensitivity, specificity, positive predictive value (PPV), and negative predictive value (NPV) were calculated by each reader (R1, R2) for dsMRI and mpMRI. Analysis was performed on a per-patient basis, categorizing a patient as a true positive when the targeted biopsy (or any biopsy in the same quadrant of the PI-RADS v2.1 map) corresponded to the index lesion, i.e., to the csPCa with the highest ISUP grade among all the biopsy cores. In the case of csPCas at different prostate locations showing the same ISUP grade, a case was assessed as a true positive if at least one of these foci were detected by targeted biopsy. Patients with csPCa undetected by mpMRI were categorized as false-negative cases, while patients with no positive biopsy and a suspicious mpMRI observation that did not correspond to csPCa were categorized as false-positives. For the purpose of analysis, csPCa was classified according to the NCCN guidelines, version 2.2019 [14] (Appendix A). Low-risk and favorable intermediate-risk PCa were considered as clinically insignificant PCa. Difference in sensitivity between dsMRI and mpMRI was assessed with the McNemar test.

Clinical utility of dsMRI and mpMRI was investigated with a decision-analysis model [18], in which the net benefit of performing these examinations was plotted against risk threshold of having csPCa, along with the treat-none and treat-all strategies, which were assessed as referring all candidate patients for AS or active treatment, respectively. Net benefit was calculated as the difference between the proportion of true positive (TP) cases and the proportion of false positive (FP) cases, with the latter adjusted for the ratio of risk threshold to 1 minus risk threshold. For the purpose of analysis, we included R1 and R2 in the model, using a risk-threshold probability of 10%, 15%, and 20%. Decision analysis was complemented with a receiver operating characteristic (ROC) analysis to calculate the area under the curve (AUC) of each model as a summary measure of diagnostic accuracy. The AUCs were compared with the DeLong method.

Finally, the inter-reader agreement between R1 and R2 in assigning the PI-RADS v2.1 category was evaluated with the Cohen’s kappa statistic. Agreement was considered to be poor when k was less than 0.20, fair when k ranged from 0.21 to 0.40, moderate when k ranged from 0.41 to 0.60, good when k ranged from 0.61 to 0.80, and very good when k was greater than 0.80 [19].

Significance for all tests was set at *p* < 0.05. Analysis were performed using SPSS version 20 (IBM, Armonk, NY, USA) and MedCalc Software version 18.11.6 (Ostend, Belgium).

## 3. Results

### 3.1. Patients’ Characteristics and Results of mpMRI and dsMRI Readings

The final population included 54 men with low-risk PCa according to clinical NCCN-based classification, whose confirmatory biopsies are shown in Table 1. The median time between PCa diagnosis and mpMRI was 6 months (5–10). The reclassification rate was 22.2% (95%CI 11.4–38.8), i.e., 12/54 patients were found to have csPCa, indicating a shift to active treatment. Table 2 shows the results of the dsMRI and mpMRI for R1 and R2. Overall, the dsMRI was positive (PI-RADS ≥3) in 39/54 cases (72.2%) and 41/54 cases (75.9%) for R1 and R2, respectively, while the mpMRI was positive in 40/54 cases (74.1%) and 42/54 cases (77.8%) for R1 and R2, respectively. Using the mpMRI, there was a decrease in PI-RADS 3 assignments for both readers, particularly R2. Notably, most of the PI-RADS 3 assigned with dsMRI were false positives, even when upgraded to PI-RADS 4 with mpMRI (Appendix A).

### 3.2. Inter-Observer Agreement

The inter-observer-agreement values for the PI-RADS category were moderate (k = 0.53; 95%CI 0.29–0.78) and good (k = 0.63; 95%CI 0.45–0.82), respectively, for the dsMRI and mpMRI.

### 3.3. Accuracy and Clinical Utility of mpMRI and dsMRI in Assessing csPCa

Table 3 summarizes the diagnostic performance of the dsMRI and mpMRI. The mpMRI showed greater sensitivity than the dsMRI, although not to a significant extent. Indeed, the mpMRI found one additional TZ csPCa for both R1 and R2, who scored it PI-RADS 1 in the transverse T2-weighted plane and PI-RADS 3 in the sagittal and coronal plane (Figure 1). The dsMRI and mpMRI showed comparable NPV, regardless of the reader. On the ROC analysis, the mpMRI showed a higher AUC than the dsMRI (Figure 2), although not to a statistically significant extent (Table 3). Figure 3 shows a case of concordance between the dsMRI and the mpMRI. The results of the decision analysis are shown in Table 4 and Figure 4. At any risk threshold, the mpMRI showed a greater net benefit than the dsMRI for both R1 and R2. The net benefit achieved by R1 when using the mpMRI was higher than the that provided by the treat-all strategy, which entailed shifting all the patients to active treatment.

## 4. Discussion

The mpMRI had a higher sensitivity and AUC than the dsMRI for assessing csPCa before starting the AS protocol. Although these results did not reach statistical significance difference, they were obtained regardless of the reader. The findings are in line with those of previous studies showing comparable results between MRI sequences, although in different clinical scenarios [12,13,14]. When using decision analysis to translate the diagnostic accuracy into clinical utility, the mpMRI showed a greater net benefit than the dsMRI and the treat-all approach (i.e., referring all patients to active treatment), regardless of the reader. The net benefit was higher at all of the risk thresholds, which we set to up to 20% (1/4 men), i.e., at the previously reported upper rate of patient reclassification on confirmatory biopsy [20,21,22]. We assumed this threshold would reasonably match the upper limit of risk at which a caregiver would weight the relative harm caused by missing csPCa as much greater than the harm caused by unnecessary active treatment.

Notably, the net benefit expresses the expected utility, irrespective of statistical significance or confidence-interval values [23]. In this light, the standard MRI protocol had the potential to be the most effective strategy with which to balance the shift from AS to active treatment while minimizing the potential side effects of overtreating patients. This is further emphasized by a lower net benefit for dsMRI than the treat-all strategy for the less experienced reader (R2). Even providing comparable sensitivity and AUC, the mpMRI should not be replaced by the dsMRI as a tool with which to guide confirmatory biopsies, especially for inexperienced readers. Furthermore, the mpMRI provided greater inter-reader agreement in the PI-RADS assessment than the dsMRI.

For each reader, the mpMRI detected the same csPCa missed by the dsMRI. This consisted of a TZ cancer whose PI-RADS categorization was better achieved on the sagittal and coronal planes of T2WI (the dominant sequence for assessing TZ cancers), which are omitted in dsMRI. This finding was in line with the previous observation that omitting these planes could make the allocation of lesions in the prostate areas more difficult, leading to an increase in PI-RADS 3 assignments [13]. Our results suggest the higher sensitivity of the standard protocol. This is due its multiplanarity rather than the use of contrast medium. Overall, the use of contrast medium did not affect cancer detection, nor did it improve the PI-RADS categorization. This is in accordance with a previous observation, according to which DCE plays a marginal role and is sometimes misleading [24].

Indeed, bpMRI preserves the multiplanarity of T2-weighted imaging while avoiding DCE. This might be used to highlight the advantages that can be expected from abbreviated protocols during patient counseling. Abbreviated protocols may indeed improve patients’ accessibility to and tolerability for MRI. This in turn could translate into better compliance with AS programs based on serial MRI examinations. This hypothesis should be tested in further studies.

We observed low PPV values, which were in line with previous data reported in a per-patient basis systematic review and metanalysis by Moldovan [25]. Several factors could explain the low PPV in our series: firstly, the low prevalence of csPCa in the whole population (22%); secondly, the category 3 [26], which was inevitably included based on the currently recommended threshold for the indication of the performance of biopsy (PI-RADS ≥3) [27]. Thirdly, the potential overrating of positive MRI results by the readers’ awareness of the clinical setting of the PCa patients [24]. This finding of low ppv values will alter the need to recommend only relevant biopsy targets for the purpose of AS MRI protocols.

For the small lesions, we cannot exclude targeting errors, since the biopsies were performed under cognitive guidance rather than through the fusion technique. On the other hand, the relevance of low per-patient PPV is questionable, as all the men were inevitably referred to biopsy in this particular clinical setting. More importantly, we observed high per-patient NPV, which is a better indicator of MRI effectiveness as a tool to avoid unnecessary biopsies and reduce the rate of indolent-cancer detection. High NPV supports the hypothesis that low-risk patients with a negative MRI might avoid and/or delay confirmatory biopsy, although a non-negligible percentage of patients with negative MRI may still have an occult csPCa [28]. The absence of cancer on confirmatory biopsy might predict a good prognosis with a lower risk of progression during AS [29]. With the introduction of mpMRI at the outset of AS, the risk of understaging at inclusion decreases and the use of per-protocol confirmatory biopsy might become less stringent, although not redundant [30,31].

Our study has some limitations, such as its retrospective nature and the relatively small number of patients included, the readers’ awareness of the cancer history, and the unavailability of MRI-guided fusion biopsy at the time when the study was conducted. Fortunately, recent evidence showed that software-based MRI-target trasperineal prostate biopsy does not offer a clear advantage compared to cognitive fusion biopsies [32]; the positive biopsies were not stratified according to random and MRI cognitive biopsy. Finally, selection bias could not be excluded, since all the patients had previously undergone a first biopsy, with a low PCa tumor volume detected.

## 5. Conclusions

The dsMRI is a useful tool with similar diagnostic performance to mpMRI for patients who are candidates for AS. However, mpMRI offers a greater net benefit than dsMRI in terms of clinical utility and diagnostic accuracy. Therefore, abbreviated dsMRI may be of potential value in clinical practice for more experienced radiologists due to dsMRI’s inadequate characterization of transitional-zone and PI-RADS 3 lesions.

## Figures and Tables

**Figure 1 diagnostics-13-00578-f001:**
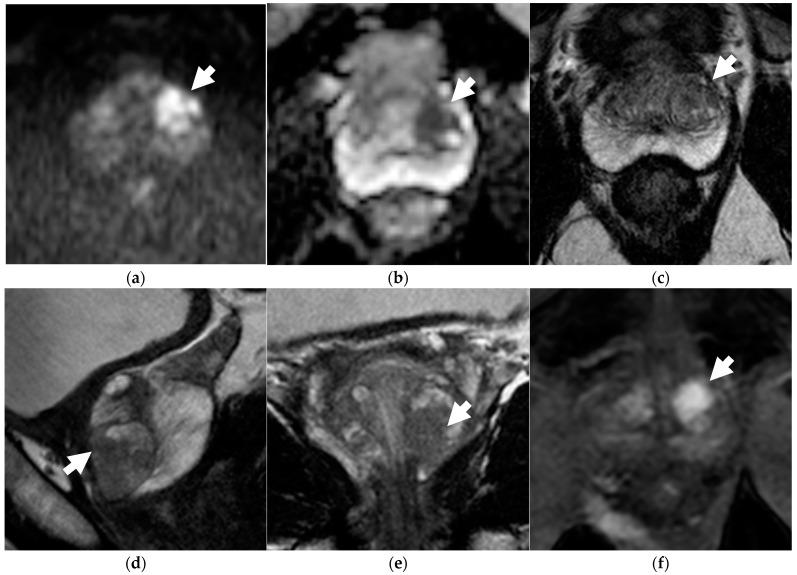
Discordance between dsMRI and mpMRI in a 66-year-old patient with unfavorable intermediate-risk prostate cancer (PSA level 6.30 ng/mL, clinical stage T1c, ISUP grade 2 on targeted biopsy). When interpreting dsMRI, both readers found an observation in the left-posterior-transition zone, showing restricted diffusion, i.e., hyperintensity on b = 2000 s/mm^2^ image (arrow in **a**), and marked hypointensity on the ADC map (arrow in **b**). The observation was categorized as PI-RADS 1 because it corresponded to a round, encapsulated nodule on axial T2WI (arrow in **c**) (typical nodule according to PI-RADS v2.1). When interpreting mpMRI, both readers categorized the finding as PI-RADS 3 on T2WI, because additional sagittal (**d**) and coronal (**e**) T2WI planes better defined an intra-nodular component showing moderate hypointensity with blurred margins (arrows). Due to restricted diffusion, the observation was finally upgraded to PI-RADS 4. On DCE (arrow in **f**), the lesion showed intense contrast enhancement.

**Figure 2 diagnostics-13-00578-f002:**
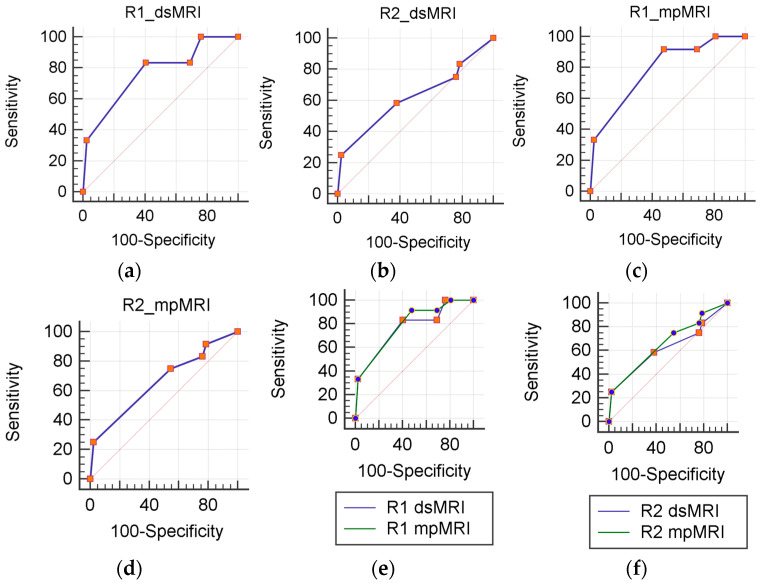
ROC analysis. dsMRI (**a**,**b**) and mpMRI (**c**,**d**) accuracy js calculated separately for each reader (R1 and R2). A statistically significant differences is not present between mpMRI and dsMRI (**e**,**f**).

**Figure 3 diagnostics-13-00578-f003:**
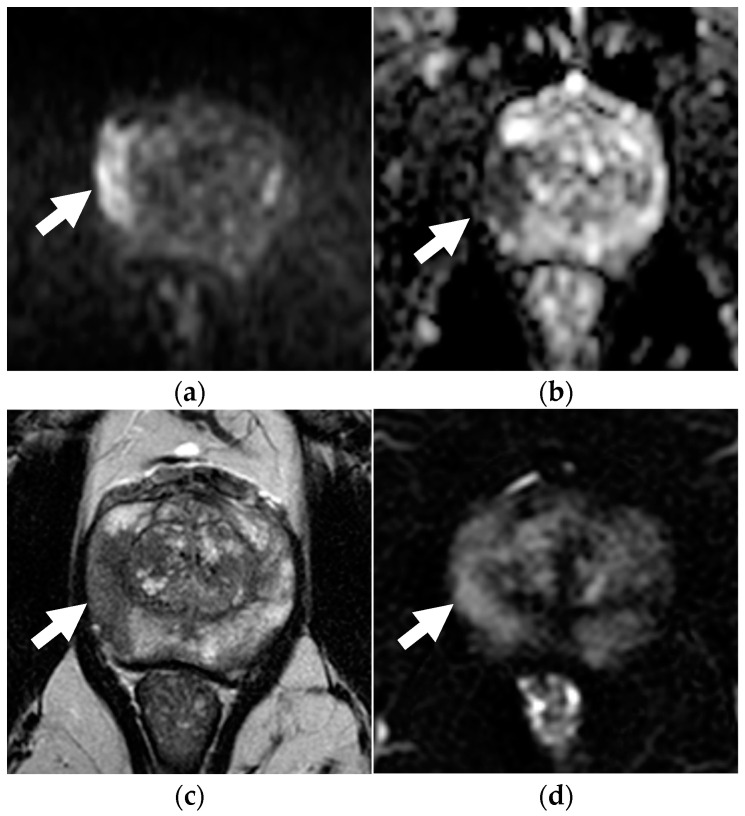
Concordance between dsMRI and mpMRI in a 72-year-old patient with unfavorable intermediate-risk prostate cancer (PSA level 5.10 ng/mL, clinical stage T1c, ISUP grade 2 with on targeted biopsy). On dsMRI, readers identified a focal zone of restricted diffusion with marked hyperintensity on b = 2000 s/mm^2^ (arrow in **a**) and hypointensity on the ADC map (arrow in **b**), categorized as PI-RADS 5. Lesion appeared as hypointense on axial T2WI (arrow in **c**). The PI-RADS categorization was the same when interpreting mpMRI as other sequences, such as DCE (arrow in **d**) added no additional information.

**Figure 4 diagnostics-13-00578-f004:**
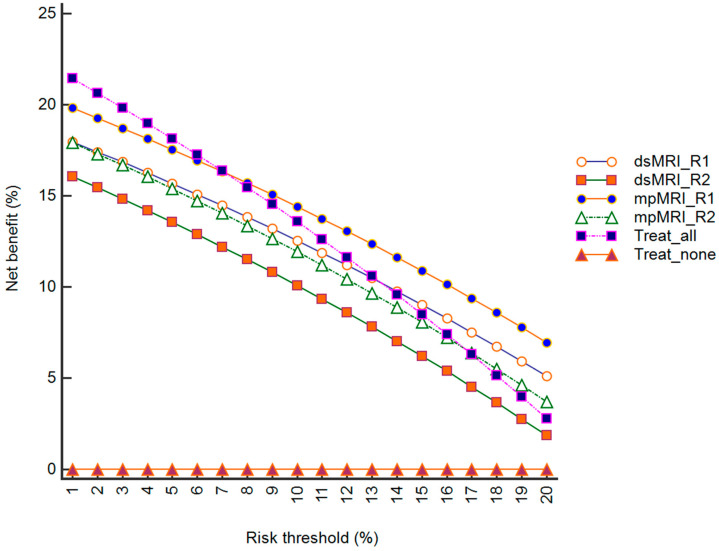
Decision-analysis curves obtained by plotting the net benefit of dsMRI, mpMRI, treat-none approach (referring all patients to AS), and treat-all approach (referring all patients to active treatment) versus risk thresholds. The net benefit of mpMRI was higher than that of dsMRI for both reader 1 (R1) and reader 2 (R2), as well as for the treat-all strategy for risks of >9% (R1) and >14% (R2).

**Table 1 diagnostics-13-00578-t001:** Characteristics of enrolled patients and pathology results of confirmatory biopsy.

Patient Characteristics	
N of patients	54
Median age, yr (IQR)	69 (69.0–73.8)
Median PSA at MRI time, ng/mL (IQR)	6 (6.0–8.3)
Median PSA density, ng/mL/mL (IQR)	0.13 (0.13–0.19)
Clinical stage, n (%)	
cT1c	41/54 (75.9%)
cT2a	13/54 (24.1%)
**Saturation-biopsy results**	
ISUP grade, at saturation biopsy, n (%)	
I	36/42 (85.7%)
II	5/42 (11.9%)
IV	1/42 (2.3%)
NCCN risk-category groups. Reclassification after biopsy, n (%)	
Low	42/54 (77.8%)
Unfavorable–Intermediate	11/54 (20.3%)
High	1/54 (1.9%)

IQR = interquartile range; PSA = prostate-specific antigen; cT = clinical T stage; ISUP = International Society of Urological Pathology; NCCN = National Comprehensive Cancer Network.

**Table 2 diagnostics-13-00578-t002:** Distribution of prostate-imaging reporting and data system (PI-RADS) assignments provided by reader 1 (R1) and reader (R2). Assignments are reported as the highest on a per-patient basis in case of multiple observations.

MRI before Confirmatory Biopsy		
Median Prostate Volume at MRI, cc (IQR)	45 (45.0–57.75)	
PI-RADS (R1) n, (%)	dsMRI	mpMRI
1	10 (18.5)	8 (14.8)
2	5 (9.3)	6 (11.1)
3	12 (22.2)	9 (16.7)
4	22 (40.7)	26 (48.1)
5	5 (9.3)	5 (9.3)
PI-RADS (R2) n, (%)	dsMRI	mpMRI
1	11 (20.4)	10 (18.5)
2	2 (3.7)	2 (3.7)
3	18 (33.3)	10 (18.5)
4	19 (35.2)	28 (51.9)
5	4 (7.4)	4 (7.4)

IQR = interquartile range.

**Table 3 diagnostics-13-00578-t003:** Accuracy of diagnosis of clinically significant prostate cancer with dual-sequence magnetic resonance imaging (dsMRI) and multiparametric MRI (mpMRI). Numbers in brackets are 95% confidence intervals. R1 = reader 1; R2 = reader 2; AUC = area under the curve.

		TP/n	FP/n	Net Benefit (%)		
	Pt			Risk Treshold 10%	Risk Treshold 15%	Risk Treshold 20%
R1	dsMRI	0.19	0.54	12.5	9.0	5.1
	mpMRI	0.20	0.54	14.4	10.9	7.0
R2	dsMRI	0.17	0.59	10.1	6.2	1.9
	mpMRI	0.19	0.59	11.9	8.1	3.7
	Treat All (shift to active treatment)	0.22	0.78	13.6	8.5	2.8
	Treat None (continue with surveillance)	/	/	0	0	0

**Table 4 diagnostics-13-00578-t004:** Results of decision analysis. The unit of net benefit is true positives (e.g., net benefit 7.0% means 7 patients were identified with csPCa per 100). TP: true positive, FP: false positive.

	R1			R2		
	dsMRI	mpMRI	*p*	dsMRI	mpMRI	*p*
TRUE-POSITIVE	10	11	-	9	10	-
FALSE-NEGATIVE	2	1	-	3	2	-
TRUE-NEGATIVE	13	13	-	10	10	-
FALSE-POSITIVE	29	29	-	32	32	-
SENSITIVITY	83.3% (52–98)	91.7% (62–100)	1.0	75.0% (43–95)	83.3% (52–98)	1.0
SPECIFICITY	31.0% (18–47)	31.0% (18–47)	-	23.8% (12–40)	23.8% (12–40)	-
PPV	25.6% (13–42)	27.5% (15–44)	-	22.0% (11–38)	23.8% (12–39)	-
NPV	86.7% (60–98)	92.9% (66–99)	-	76.9% (46–95)	83.3% (52–98)	-
AUC	0.77 (0.63–0.87)	0.79 (0.66–0.89)	0.65	0.62 (0.48–0.75)	0.66 (0.52–0.78)	0.52

## Data Availability

The data presented in this study are available on request from the corresponding author. The data are not publicly available due to ethical restrictions.

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
