# Peer review of "Abbreviated Versus Multiparametric Prostate MRI in Active Surveillance for Prostate-Cancer Patients: Comparison of Accuracy and Clinical Utility as a Decisional Tool"

_diagnostics, 2023, doi:10.3390/diagnostics13040578_

Round 1

Reviewer 1 Report

This is a retrospective study assessing the diagnostic impact of dual-sequence MRI in a group of patients undergoing active surveillance for low-risk prostate cancer. There is a big interest in abbreviated MRI modalities, due to significant examination time and cost reductions. However, it is not yet clear if the diagnostic performance of these techniques is comparable to that of multiparametric MRI, especially in the hands of an inexperienced radiologist. 

Therefore this work could be of interest for publication. 

I have some concerns:

1. Methods: The authors mention that the study obtained an ethical approval by the institutional board/ethical committee. However they do not provide the ethical approval number. Although the study is retsrospective, an ethical approval is still required as patient data are being circulated and published.  

2. Methods: Retrospective design and prospective maintained dataset cannot go together. The study is either prospective or retrospective.  

3. Methods: The authors should specify the temporal  cut-off between PCa diagnosis and mpMRI

4. Methods: The authors should specify the reason for performing saturation biopsies. Did they use a standard (eg Ginsburg) scheme? Saturation biopsy could influence the detection rates. 

5. Methods: The authors should define the TP and FP abbreviations 

6. Results: The authors should better specify and evaluate positive findings. How many patients had positive saturation and how many had positive targeted cores only?

7. Results:  Figure 3 legend is double 

8. References: some of them are of low impact and should be removed or replaced (eg no 5, 7, 20, 26)

9. Language: the text needs linguistic improvements by a native speaker or a language editing software  

Reviewer 2 Report

Dear authors.

Thank you for submitting this interesting data showing the comparison between the dsMRI and the mpMRI in patients under active surveillance by two radiologists with different lengths of experience.

It is known from the Promis study that mpMRI has advantages over systematic biopsy for the detection of a clinically significant PCa (Gleason >=3+4). In the active surveillance setting, retrospective studies showed that a positive mpMRI before radical prostatectomy is associated with a higher risk of upgrading and with a confirmatory biopsy in patients in active surveillance with a higher risk of reclassification. A negative mpMRI does not preclude upgrading or reclassification. This led to the conclusion that targeted and systematic biopsies should be used for optimal detection of a clinically significant PCa. In doi:10.1148/radiol.2017170129 it could be shown that the biparametric MRI offers a similar diagnostic accuracy as the mpMRI. Strikingly, as in the results of the present study by Zattoni et al. is the high false positive rate. Another weakness of this study is the inability to perform fusion biopsies, which has already been noted.

Figure 2 (ROC curves) is missing in my version and the numbering of the figures has gotten mixed up. In addition, it is better to use the color map of the decision curve analysis. Please remove the black and white image.

Round 2

Reviewer 1 Report

I have no further comments. The study can be accepted in its present form.